# LoRAQuant: Mixed-Precision Quantization of LoRA to Ultra-Low Bits

## Abstract

Low-Rank Adaptation (LoRA) has become a popular technique for parameter-efficient fine-tuning of large language models (LLMs). In many real-world scenarios, multiple adapters are loaded simultaneously to enable LLM customization for personalized user experiences or to support a diverse range of tasks. Although each adapter is lightweight in isolation, their aggregate cost becomes substantial at scale. To address this, we propose LoRAQuant, a mixed-precision post-training quantization method tailored to LoRA. Specifically, LoRAQuant reparameterizes each adapter by singular value decomposition (SVD) to concentrate the most important information into specific rows and columns. This makes it possible to quantize the important components to higher precision, while quantizing the rest to ultra-low bitwidth. We conduct comprehensive experiments with LLaMA 2-7B, LLaMA 2-13B, and Mistral 7B models on mathematical reasoning, coding, and summarization tasks. Results show that our LoRAQuant uses significantly lower bits than other quantization methods, but achieves comparable or even higher performance.[1]

## 1 Introduction

Large Language Models (LLMs) have achieved remarkable performance across a wide range of natural language tasks (Ouyang et al., 2022; Wang et al., 2022; Zhao et al., 2023), but fine-tuning LLMs for new applications remains computationally and memory intensive. To address this challenge, **lo**w-**ra**nk adaptation (LoRA; Hu et al., 2022) has emerged as a widely adopted method for parameter-efficient fine-tuning. LoRA introduces small, task-specific low-rank matrices, and during the adaptation, only these low-rank matrices are trained while the base model is frozen.

An increasingly important use case of LoRA is LLM customization, as LLM providers (e.g., OpenAI and Google) allow users to personalize their own LLMs (OpenAI, 2025; Google Cloud, 2025). This could result in hundreds of millions of customized LLMs addressing diverse tasks, domains, and users. This imposes significant challenges of storing and using these massive customized LLMs.

A straightforward attempt to solve these challenges is to freeze the base LLM and train a separate adapter for each customization (Sheng et al., 2024). During deployment, multiple LoRAs are often loaded simultaneously due to parallel user requests, and thus, the memory footprint of LoRAs becomes a concern, especially if the GPU memory is small. This is because, although each individual LoRA is relatively lightweight, the cumulative GPU memory consumption for loading many adapters can become significant.

To scale multi-LoRA systems, Gabrielsson et al. (2024) propose a compression technique that clusters different LoRAs and enables representation sharing within each cluster. However, a key limitation of their method is that it requires recomputing the shared parameters whenever a new adapter is added. Additionally, their evaluation is primarily conducted on relatively simple tasks where the base LLM already performs well and their approach struggles on more challenging tasks.

In this paper, we propose LoRAQuant, a post-training LoRA quantization method for LLM customization. Quantization is a well-established technique for compressing neural networks, and is able to substantially reduce the parameter space without degrading performance much (Frantar et al.,

---

[1]Our code is released anonymously at: `https://github.com/Anonymous890920/LoRAQuant`

2023; Lin et al., 2024). Despite this progress, the quantization of LoRA has received little attention in the literature, compared with quantizing full LLMs. Although existing quantization methods can be directly applied to LoRA weights, they overlook the unique low-rank structure of LoRA and perform poorly at ultra-low precisions (e.g., 1–2 bits).

In our work, we observe that a LoRA is a product of two low-rank matrices, which can be easily split into multiple *lower*-rank adapters (sub-LoRAs). By reparametrizing the LoRA by singular value decomposition (SVD), we can perform mixed-precision quantization for different sub-LoRAs: more precisions for more important SVD dimensions and fewer precisions for less important ones. With our approach, we are able to retain high performance of LoRA while reducing the memory space to a large extent.

In the experiments, we evaluate our method by training adapters on three representative models LLaMA2-7B, LLaMA2-13B (Touvron et al., 2023), and Mistral-7B (Jiang et al., 2023), across diverse tasks including mathematical reasoning, code generation, and summarization. Our results demonstrate that LORAQUANT achieves competitive performance even under ultra-low bitwidth for LoRA (less than 2 bits on average).

## 2 RELATED WORK

**Model quantization.** Quantization has become an important technique for reducing the memory footprint of LLMs, enabling efficient deployment without a significant loss in accuracy. A variety of post-training methods have been proposed to compress full-precision weights into lower-bit representations. For instance, GPTQ (Frantar et al., 2023) leverages second-order information to minimize quantization error, while AWQ (Lin et al., 2024) incorporates activation statistics to guide weight quantization. InvarExplore (Wen et al., 2025) leverages model invariances (namely, rotation, scaling, and permutation) to make weights perform better after quantization. SVDQuant (Li et al., 2025) performs quantization on a weight matrix, but adds a full-precision SVD with a few dimensions to improve performance. Our work differs from SVDQuant (although also using SVD) in that we focus on LoRA quantization and use the SVD decomposition to split the LoRA into two sub-LoRAs: one containing more important information and the other less important information.

Although the above approaches primarily target moderate quantization (e.g., 3–8 bits), an even more extreme direction is binarization, where weights are restricted to two values (Rastegari et al., 2016). Pure binary quantization usually achieves very low performance, and researchers have proposed mixed-precision methods where some weights are binarized while others are kept in high precision. For example, PB-LLM (Shang et al., 2024) uses an additional bit to indicate whether a weight is binarized or not, which unfortunately offsets the memory saved. BiLLM (Huang et al., 2024) also adopts mixed-precision quantization, but restricts the binarization to certain column of the weight matrix. However, BiLLM still requires an additional indicator bit, as it employs a split binarization strategy in which the weights are divided into two groups and binarized separately. The extra bit is needed to indicate the group membership of each weight.

**Low-rank adapter (LoRA).** LoRA (Hu et al., 2022) has become a widely adopted approach for parameter-efficient fine-tuning of LLMs. Building on this idea, several extensions aim to further reduce memory overhead or improve training effectiveness. Meng et al. (2024) initialize LoRA with the SVD of the base weight matrix rather than random values, providing a stronger starting point for optimization. Zhang et al. (2023) dynamically adjust the rank of LoRA during training, allocating higher ranks to more important layers. Hao et al. (2024) and Lialin et al. (2024) address the limitation that LoRA updates may remain low-rank by iteratively merging and resampling adapters during training. Kopiczko et al. (2024) propose sharing LoRA weights across layers to reduce memory.

LoRA has also been applied to fine-tuning quantized models. QLoRA (Dettmers et al., 2023) freezes the quantized base model while training an add-on LoRA in full precision. LoftQ (Li et al., 2024) and ApiQ (Liao et al., 2024) improve QLoRA adapter initialization by choosing parameters that reduce quantization errors instead of random values. QA-LoRA (Xu et al., 2024) extends QLoRA by ensuring that the adapter weights remain easily quantizable even after being merged with the original weights at the cost of reducing the representational capacity of the adapters during training.

Note that our LORAQUANT is completely different from QLoRA and its variants, despite similar names. QLoRA uses a full-precision LoRA to fine-tune a quantized model, as the latter cannot

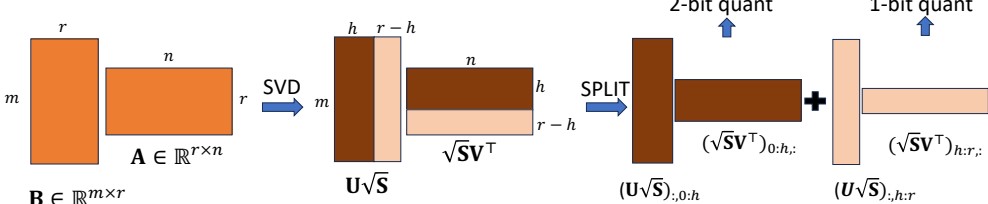

Figure 1: Overview of our LORAQUANT method.

be easily trained. Our work focuses on the post-training quantization of LoRAs. It is arguable that LoRA is already a parameter-efficient method, but with increasingly many LoRAs for LLM customization, the aggregated cost can be substantial.

In the context of serving multiple LoRAs simultaneously, prior work has focused on both hardware improvement and LoRA compression. S-LoRA (Sheng et al., 2024) proposes a batched inference strategy to improve throughput when handling concurrent LoRA requests. Punica (Chen et al., 2024) implements a customized GPU kernel, Segmented Gather Matrix-Vector Multiplication (SGMV), which enables efficient batching across heterogeneous LoRAs. Complementary to these approaches, Gabrielsson et al. (2024) reduce the memory footprint by weight sharing among a cluster of LoRAs and assigning a few additional parameters for each LoRA. However, our experiments will show that this approach does not perform well in sophisticated tasks such as math reasoning and coding (§4.2).

## 3 OUR LORAQUANT METHOD

In our LORAQUANT method, we decompose a LoRA into two lower-rank adapters, called sub-LoRAs (§3.1). Then, we allocate slightly higher precision to the more important sub-LoRA , and perform extreme 1-bit quantization for the less important sub-LoRA (§3.2). To further mitigate quantization error, we also apply gradient-based optimization before quantization (§3.3). An overview of the approach is shown in Fig. 1, and the step-by-step procedure is given in Algs. 1 and 2.

### 3.1 SPLITTING A LORA INTO SUB-LORAS BY SVD

For an update of neural network weights $\mathbf{W} \leftarrow \mathbf{W} + \Delta\mathbf{W}$, a low-rank adapter (LoRA; Hu et al., 2022) learns two low-rank matrices $\mathbf{B} \in \mathbb{R}^{m \times r}$ and $\mathbf{A} \in \mathbb{R}^{r \times n}$ to approximate $\Delta\mathbf{W}$, where $r \ll \min(m, n)$. In other words, the update becomes $\mathbf{W} \leftarrow \mathbf{W} + \mathbf{BA}$.

To enable mixed-precision quantization, we split the LoRA into sub-LoRAs, and in particular, we have two sub-LoRAs in our experiments. In other words, we need to find sub-LoRAs, namely, $\mathbf{B}_h\mathbf{A}_h$ and $\mathbf{B}_l\mathbf{A}_l$, such that $\mathbf{B}_h\mathbf{A}_h + \mathbf{B}_l\mathbf{A}_l = \mathbf{BA}$. Later, $\mathbf{B}_h$ and $\mathbf{A}_h$ will be quantized with higher precisions, whereas $\mathbf{B}_l$ and $\mathbf{A}_l$ will be quantized with lower precisions.

The key challenge is to determine how to perform the split to suit mixed-precision quantization. Intuitively, the more important components should be allocated to the higher-precision sub-LoRA, while the less important components can be assigned to the lower-precision sub-LoRA. A naïve strategy is to split $\mathbf{B}$ and $\mathbf{A}$ by selecting certain columns and rows based on weight norms or quantization error. However, information is usually scattered over rows and columns of neural weights, so such an approach is less effective.

Instead, we propose to reparameterize a low-rank adapter $\mathbf{BA}$ into an equivalent factorization $\mathbf{B}'\mathbf{A}'$ such that $\mathbf{BA} = \mathbf{B}'\mathbf{A}'$, where important information is concentrated in specific rows and columns of $\mathbf{B}'$ and $\mathbf{A}'$. This will better suit our mixed-precision quantization scheme.

To accomplish this, we apply the singular value decomposition (SVD) to the original adapter $\mathbf{BA}$:

$$\mathbf{BA} = \mathbf{USV}^\top, \tag{1}$$

where $\mathbf{U} \in \mathbb{R}^{m \times r}$ and $\mathbf{V} \in \mathbb{R}^{n \times r}$ are orthonormal matrices, and $\mathbf{S} \in \mathbb{R}^{r \times r}$ is a diagonal matrix containing the singular values in descending order. Here, the SVD is truncated to $r$ ranks, since $\mathbf{BA}$ has at most $r$ ranks. We reparameterize the LoRA by

$$\mathbf{B}' = \mathbf{US}^{1/2}, \quad \mathbf{A}' = \mathbf{S}^{1/2}\mathbf{V}^\top, \tag{2}$$

where $\mathbf{S}^{1/2}$ is the diagonal matrix whose entries are the square roots of the singular values. It is easy to verify $\mathbf{B}'\mathbf{A}' = (\mathbf{US}^{1/2})(\mathbf{S}^{1/2}\mathbf{V}^\top) = \mathbf{USV}^\top = \mathbf{BA}$.

This reparameterization ranks the importance of each component (i.e., each column of $\mathbf{B}'$ and corresponding row of $\mathbf{A}'$) by the magnitude of its associated singular value. We retain the top-$h$ components in higher precision and quantize the remaining $r - h$ components using a lower bitwidth.

Formally, the more important sub-LoRA is given by:

$$\mathbf{B}_h = (\mathbf{US}^{1/2})_{[:,\, 0:h]} \quad \mathbf{A}_h = (\mathbf{S}^{1/2}\mathbf{V}^\top)_{[0:h,\, :]} \tag{3}$$

and the less important sub-LoRA is given by:

$$\mathbf{B}_l = (\mathbf{US}^{1/2})_{[:,\, h:r]}, \quad \mathbf{A}_l = (\mathbf{S}^{1/2}\mathbf{V}^\top)_{[h:r,\, :]}, \tag{4}$$

where the subscript $[\cdot, \cdot]$ chooses certain columns and rows. We can easily see that $\mathbf{B}_h\mathbf{A}_h + \mathbf{B}_l\mathbf{A}_l = \mathbf{B}'\mathbf{A}' = \mathbf{BA}$, showing that our transformations do not change the functionality of the LoRA.

**Determining the ratio of two sub-LoRAs.** We employ a dynamic strategy to compute $h$ for each adapter, based on the coverage of total variance of that adapter. Specifically, we introduce a ratio hyperparameter $\rho \in (0, 1]$, which specifies the minimum ratio of total variance that must be preserved. Given the singular values $s_1, s_2, \cdots, s_r$ (sorted from largest to smallest) of an adapter, we compute $h$ as the smallest integer satisfying:

$$\frac{\sum_{i=1}^h s_i^2}{\sum_{i=1}^r s_i^2} \geq \rho \tag{5}$$

This ensures that the top-$h$ singular directions collectively explain at least $\rho \times 100\%$ of the variance in the adapter product. Compared with directly setting $h$ as a hyperparameter, our strategy allows us to adaptively allocate more precision to layers that require a greater number of singular components to preserve their representational capacity.

### 3.2 QUANTIZATION METHODS

After splitting a LoRA into two sub-LoRAs, we perform mixed-precision quantization. For the more important sub-LoRA $\mathbf{B}_h\mathbf{A}_h$, we use round-to-nearest quantization (RTN; Jacob et al., 2018) with higher precisions (e.g., 2 bits for a weight). For the less important sub-LoRA $\mathbf{B}_l\mathbf{A}_l$, we use binary quantization with only one bit for extreme compression. Profoundly, our mixed-precision quantization with 2 bits and 1 bit allows us to achieve less than 2 bits on average, which is a setting where existing methods cannot perform well (Shang et al., 2024; Huang et al., 2024).

In the rest of this part, we present RTN and binary quantization methods in detail. Notice that the quantization applies to both $\mathbf{A}_\bullet$ and $\mathbf{B}_\bullet$ (where the subscript $\bullet$ refers to either $h$ or $l$). We take $\mathbf{A}_\bullet$ as an example in the following presentation.

**RTN quantization for the more important sub-LoRA.** We employ the widely used round-to-nearest (RTN) method to quantize $\mathbf{B}_h$ and $\mathbf{A}_h$. Let us consider $\mathbf{A}_h$. RTN maps each real-valued weight to the closest integer value, but with a scaling factor $S$ and a zero-point offset $Z$ to cover the range of the weight values. Formally, the integer weights after quantization, denoted by $\bar{\mathbf{A}}_h$, are

$$\bar{\mathbf{A}}_h = Q_{\text{RTN}}(\mathbf{A}_h) = \text{round}\left(\frac{\mathbf{A}_h}{S}\right) + Z \tag{6}$$

In RTN, the largest real value in $\mathbf{A}$ is mapped to the largest representable integer, and the smallest value is mapped to the smallest representable integer. Based on this, $S$ and $Z$ are given by

$$S = \frac{\max(\mathbf{A}_h) - \min(\mathbf{A}_h)}{q_{\max} - q_{\min}}, \qquad Z = \text{round}\left(q_{\min} - \frac{\min(\mathbf{A}_h)}{S}\right), \tag{7}$$

where $q_{\min}$ and $q_{\max}$ denote the minimum and maximum integers representable under the chosen bitwidth. When using the weights, we perform dequantization as $D_{\text{RTN}}(\bar{\mathbf{A}}_h) = S \cdot (\bar{\mathbf{A}}_h - Z)$.

In implementation, we apply group-wise quantization (Jacob et al., 2018), i.e., the above procedure is performed on a group of contiguous weights instead of the entire matrix. At the cost of introducing more scaling and offset values for fine-grained treatment, this reduces quantization error.

**Binary quantization for the less important sub-LoRA.** We perform binary quantization on the less important sub-LoRA $\mathbf{B}_l\mathbf{A}_l$ for extreme compression. The classic RTN method is unsuitable

in the 1-bit setting, as it maps weights to either $\{0, +S\}$ or $\{0, -S\}$ and cause many weights to collapse to zero due to Eqn. (6), during which significant information is lost.

Instead, we adopt the binary quantization method in Rastegari et al. (2016), which maps values to $\{-S, +S\}$ so as to preserve more representational capacity. In other words, the quantization and dequantization processes are simply given by

$$\bar{\mathbf{A}}_l = Q_{\text{bin}}(\mathbf{A}_l) = \text{sign}(\mathbf{A}_l), \qquad D_{\text{bin}}(\bar{\mathbf{A}}_l) = S \cdot \bar{\mathbf{A}}_l, \tag{8}$$

where $\text{sign}(x) = 1$ if $x \geq 0$, or $-1$ otherwise. The scaling factor $S$ is set as $S = \frac{1}{n}\|\mathbf{A}_l\|_1$, which is shown to minimize the Frobenius norm between the original weights $\mathbf{A}_l$ and the reconstructed one $D_{\text{bin}}(\bar{\mathbf{A}}_l)$ (Rastegari et al., 2016). Like RTN, we also use group-wise quantization, where the scaling factor is computed separately within each group of weights.

### 3.3 OPTIMIZING THE SUB-LoRAs BY STRAIGHT-THROUGH GRADIENT DESCENT

The quantization error can be further reduced by an optimization process. This is typically accomplished by searching for an optimal reparameterization $\mathbf{B}^*\mathbf{A}^*$ of the original LoRA $\mathbf{B}\mathbf{A}$ such that the quantization error is minimized. In our approach, we perform optimization on each column of $\mathbf{B}_\bullet$ and its corresponding row of $\mathbf{A}_\bullet$, one pair at a time. This is because we have performed SVD and do not want to mix the singular dimensions during joint optimization.[2]

Let $\boldsymbol{b}_i \in \mathbb{R}^m$ be the $i$th column of $\mathbf{B}_\bullet$, and $\boldsymbol{a}_i \in \mathbb{R}^n$ be the $i$th row of $\mathbf{A}_\bullet$; both $\boldsymbol{b}_i$ and $\boldsymbol{a}_i$ are column vectors. The goal is to find $\boldsymbol{b}_i^* \in \mathbb{R}^m$ and $\boldsymbol{a}_i^* \in \mathbb{R}^n$ to

$$\underset{\boldsymbol{b}_i^*, \boldsymbol{a}_i^*}{\text{minimize}} \left\| \boldsymbol{b}_i \boldsymbol{a}_i^\top - D(Q(\boldsymbol{b}_i^*))D(Q(\boldsymbol{a}_i^{*\top})) \right\|_F \tag{9}$$

where $D$ and $Q$ are one of the quantization methods in §3.2, depending on which sub-LoRA we are handling.

In other words, we reparameterize each SVD dimension such that its quantization has a smaller error. The vector $\boldsymbol{b}_i^*$ and $\boldsymbol{a}_i^*$ are initialized with $\boldsymbol{b}_i$ and $\boldsymbol{a}_i$, respectively, and are learned through gradient-based optimization. Due to the non-differentiable dequantization function, we employ the Straight-Through Estimator (STE; Bengio et al., 2013) during backpropagation, which approximates the gradient by treating the rounding function as an identity function. This allows the gradient to flow despite the non-differentiable nature of quantization.

In practice, the optimization converges within one hundred gradient steps, and thus is computationally efficient. After optimizing $\boldsymbol{b}_i^*$ and $\boldsymbol{a}_i^*$ for different $i$ values, they are put back into two new matrices $\mathbf{B}_\bullet^*$ and $\mathbf{A}_\bullet^*$ for quantization according to §3.2.

## 4 EXPERIMENTS

We provide the setting of our experiment in §4.1, and present our main result in §4.2. Further analyses and ablation studies are provided in §4.3.

### 4.1 SETTINGS

To evaluate the effectiveness of LORAQUANT, we train LoRA for three widely used open-weight language models: LLaMA 2-7B, LLaMA 2-13B (Touvron et al., 2023), and Mistral 7B (Jiang et al., 2023). We apply the LoRAs to three distinct tasks: mathematical reasoning, code generation, and summarization. For each task, we use standard benchmark datasets for evaluation. For the mathematical reasoning task, we assess performance on the GSM8K (Cobbe et al., 2021) and MATH (Hendrycks et al., 2021) datasets using the LM Evaluation Harness framework (Gao et al., 2024); we report pass@1 accuracy as the evaluation metric. For code generation, we evaluate our model on the HumanEval dataset (Chen et al., 2021) with the Code Generation LM Evaluation Harness framework (Ben Allal et al., 2022); the metric is the accuracy of generated programs, where a program is considered accurate if it passes all test cases. For summarization, we evaluate on the XSum dataset (Narayan et al., 2018) and report ROUGE-L (Lin, 2004) as the metric.

---

[2]In our pilot study, we also experimented with joint optimization and observed no noticeable difference. Nevertheless, we adopt this approach because it is more intuitive.

---

**Algorithm 1** QUANTIZELORA ($\mathbf{B}$, $\mathbf{A}$, $h$, $bits_{\text{high}}$, $bits_{\text{low}}$, $T$, $\eta$)

---

**Require:** LoRA matrices $\mathbf{B} \in \mathbb{R}^{m \times r}$, $\mathbf{A} \in \mathbb{R}^{r \times n}$, ratio $\rho$, bitwidth $bits_{\text{high}}$ and $bits_{\text{low}}$, optimization steps $T$, learning rate $\eta$

1: Compute SVD: $\mathbf{U}, \mathbf{S}, \mathbf{V}^{\top} \leftarrow \text{SVD}(\mathbf{B}\mathbf{A})$
2: $\mathbf{B}' \leftarrow \mathbf{U}\sqrt{\mathbf{S}}$    {Square root is applied element-wise}
3: $\mathbf{A}' \leftarrow \sqrt{\mathbf{S}}\mathbf{V}^{\top}$
4: Find smallest $h$ such that $\frac{\sum_{i=1}^{h} s_i^2}{\sum_{i=1}^{r} s_i^2} \geq \rho$
5: $\mathbf{B}_{\text{h}} \leftarrow$ first $h$ columns of $\mathbf{B}'$
6: $\mathbf{A}_{\text{h}} \leftarrow$ first $h$ rows of $\mathbf{A}'$
7: $\mathbf{B}_{\text{l}} \leftarrow$ last $r-h$ columns of $\mathbf{B}'$
8: $\mathbf{A}_{\text{l}} \leftarrow$ last $r-h$ rows of $\mathbf{A}'$
9: **for** $i = 0$ **to** $h-1$ **do**
10:     $\mathbf{B}_{\text{h}}[:, i], \mathbf{A}_{\text{h}}[i, :] \leftarrow \text{opt}(\mathbf{B}_{\text{h}}[:, i], \mathbf{A}_{\text{h}}[i, :], T, \eta)$
11: **end for**
12: **for** $i = 0$ **to** $r-h-1$ **do**
13:     $\mathbf{B}_{\text{l}}[:, i], \mathbf{A}_{\text{l}}[i, :] \leftarrow \text{opt}(\mathbf{B}_{\text{l}}[:, i], \mathbf{A}_{\text{l}}[i, :], T, \eta)$
14: **end for**
15: $\mathbf{B}_{\text{h}} \leftarrow \text{quantize}(\mathbf{B}_{\text{h}}, bits_{\text{high}})$, $\mathbf{A}_{\text{h}} \leftarrow \text{quantize}(\mathbf{A}_{\text{h}}, bits_{\text{high}})$
16: $\mathbf{B}_{\text{l}} \leftarrow \text{quantize}(\mathbf{B}_{\text{l}}, bits_{\text{low}})$, $\mathbf{A}_{\text{l}} \leftarrow \text{quantize}(\mathbf{A}_{\text{l}}, bits_{\text{low}})$
17: **return** $(\mathbf{B}_{\text{h}}, \mathbf{A}_{\text{h}}), (\mathbf{B}_{\text{l}}, \mathbf{A}_{\text{l}})$

---

**Algorithm 2** OPT($\mathbf{B}$, $\mathbf{A}$, $bits$, T, $\eta$)

---

**Require:** Factor matrices $\mathbf{B}$, $\mathbf{A}$, target bitwidth $bits$, optimization step $T$, learning rate $\eta$

1: $\mathbf{B}_{\text{opt}} \leftarrow \mathbf{B}$, $\mathbf{A}_{\text{opt}} \leftarrow \mathbf{A}$
2: **for** $t = 1$ **to** $T$ **do**
3:     $\mathbf{Q}_B \leftarrow \text{quantize}(\mathbf{B}_{\text{opt}}, bits)$, $\mathbf{Q}_A \leftarrow \text{quantize}(\mathbf{A}_{\text{opt}}, bits)$
4:     $\mathbf{B}_{\text{rec}} \leftarrow \text{dequantize}(\mathbf{Q}_B)$, $\mathbf{A}_{\text{rec}} \leftarrow \text{dequantize}(\mathbf{Q}_A)$
5:     $\mathcal{L} \leftarrow \|\mathbf{B}\mathbf{A} - \mathbf{B}_{\text{rec}}\mathbf{A}_{\text{rec}}\|_F$
6:     Backpropagate $\mathcal{L}$ by straight-through estimation
7:     $\mathbf{B}_{\text{opt}} \leftarrow \mathbf{B}_{\text{opt}} - \eta \cdot \nabla_{B_{\text{opt}}}\mathcal{L}$
8:     $\mathbf{A}_{\text{opt}} \leftarrow \mathbf{A}_{\text{opt}} - \eta \cdot \nabla_{A_{\text{opt}}}\mathcal{L}$
9: **end for**
10: **return** $\mathbf{B}_{\text{opt}}, \mathbf{A}_{\text{opt}}$

---

For LoRA adapters, we train them separately for each task, mimicking the scenario of LLM customization. For mathematical reasoning, we use the MetaMathQA dataset (Yu et al., 2023) for training, noticing that the training set of GSM8K and MATH is too small. For code generation, we train the LoRA on the Magicoder-Eval-100-Instruct dataset (Wei et al., 2024), since HumanEval is a test-only dataset. Such training and evaluation follow the common practice in prior work (Biderman et al., 2024; Meng et al., 2024). For summarization, we use the standard training split of the XSum dataset (Narayan et al., 2018).

In all experiments, we adopt a widely used LoRA setup (Biderman et al., 2024), where the rank is set to 16 and LoRA modules are inserted into every linear layer of the transformer. The training of LoRA follows that in the QLoRA study (Dettmers et al., 2023), where the base language model is quantized and frozen whereas the LoRA is trained in half precision (FP16). Training hyperparameters are provided in Appendix A. After training, we apply our LORAQUANT method and other quantization baselines to the LoRA weights and evaluate the performance of each approach.

### 4.2 MAIN RESULTS

We present the main results in Tab. 1, where we consider the following popular baselines for quantizing LoRA. **GPTQ** (Frantar et al., 2023) quantizes weights sequentially and adjusts the remaining weights to reduce quantization error. **RTN** and **BIN** are the standard methods introduced in §3.2. **PB-LLM** (Shang et al., 2024) performs mixed-precision quantization with some weight being bina-

| Model | # | Method | Tasks | | | | Avg Perf. | Avg Bit |
|---|---|---|---|---|---|---|---|---|
| | | | GSM8K | MATH | HumanEval | XSum | | |
| LLaMA 2-7B | 1 | **FP16** | 58.53 | 18.03 | 34.76 | 33.53 | 36.21 | 16 |
| | 2 | BIN | 28.89 | 6.74 | 20.12 | 24.07 | 19.95 | 1.13 |
| | 3 | RTN (1 bit) | 0 | 2.95 | 9.76 | 8.27 | 5.24 | 1.13 |
| | 4 | JD-Diagonal | 38.29 | 6.91 | 15.85 | 30.23 | 22.82 | 5.33 |
| | 5 | RTN (2 bits) | 49.36 | 9.94 | 29.27 | 33.23 | 30.45 | 2.14 |
| | 6 | GPTQ (2 bits) | 52.16 | 13.23 | 29.88 | 33.02 | 32.07 | 2.14 |
| | 7 | PBLLM | 50.57 | 11.20 | 28.05 | 32.42 | 30.56 | 2.83 |
| | 8 | BiLLM | 53.90 | 13.90 | 29.88 | 32.86 | 32.63 | 2.24 |
| | 9 | LoRAQuant (2@0.8) | 51.25 | 10.11 | 24.39 | 32.43 | 29.55 | **1.65** |
| | 10 | LoRAQuant (2@0.9) | 52.16 | 12.72 | 29.27 | 32.43 | 31.65 | 1.81 |
| | 11 | LoRAQuant (3@0.8) | 53.60 | 14.57 | 29.88 | 33.35 | 32.86 | 2.16 |
| | 12 | LoRAQuant (3@0.9) | **56.86** | **16.01** | **31.71** | **33.51** | **34.52** | 2.5 |
| Mistral 7B | 1 | **FP16** | 58.83 | 19.46 | 45.12 | 31.96 | 38.77 | 16 |
| | 2 | BIN | 29.26 | 9.18 | 18.90 | 8.74 | 16.52 | 1.13 |
| | 3 | RTN (1 bit) | 13.42 | 11.37 | 7.32 | 15.43 | 11.88 | 1.13 |
| | 4 | JD-Diagonal | 0 | 6.49 | 3.66 | 15.08 | 6.31 | 5.33 |
| | 5 | RTN (2 bits) | 26.08 | 5.31 | 34.15 | 31.42 | 24.24 | 2.14 |
| | 6 | GPTQ (2 bits) | 40.26 | 9.69 | 31.71 | 32.46 | 28.53 | 2.14 |
| | 7 | PBLLM | 50.04 | 17.94 | 38.41 | 33.43 | 34.96 | 2.83 |
| | 8 | BiLLM | 50.95 | 14.41 | 42.07 | 32.32 | 34.94 | 2.24 |
| | 9 | LoRAQuant (2@0.8) | 52.08 | 16.43 | 35.98 | 33.26 | 34.44 | **1.85** |
| | 10 | LoRAQuant (2@0.9) | **53.90** | 17.94 | 39.63 | **33.48** | 36.24 | 1.97 |
| | 11 | LoRAQuant (3@0.8) | 51.18 | 18.45 | **44.51** | 33.01 | 36.79 | 2.56 |
| | 12 | LoRAQuant (3@0.9) | 53.75 | **18.70** | 43.90 | 32.75 | **37.28** | 2.8 |
| LLaMA 2-13B | 1 | **FP16** | 61.79 | 18.79 | 46.34 | 35.16 | 40.52 | 16 |
| | 2 | BIN | 27.37 | 9.6 | 21.34 | 31.97 | 22.57 | 1.13 |
| | 3 | RTN (1 bit) | 0 | 4.13 | 13.41 | 1.35 | 4.72 | 1.13 |
| | 4 | JD-Diagonal | 46.17 | 8.51 | 18.91 | 33.14 | 26.68 | 5.33 |
| | 5 | RTN (2 bits) | 53.30 | 14.49 | 32.93 | 34.54 | 33.81 | 2.14 |
| | 6 | GPTQ (2 bits) | 57.77 | 15.08 | 36.59 | 34.56 | 36 | 2.14 |
| | 7 | PBLLM | 59.06 | 16.34 | 32.93 | 34.08 | 35.60 | 2.83 |
| | 8 | BiLLM | 59.89 | 17.02 | 36.59 | 34.79 | 37.07 | 2.24 |
| | 9 | LoRAQuant (2@0.8) | 56.63 | 15.08 | 30.49 | 34.05 | 34.06 | **1.65** |
| | 10 | LoRAQuant (2@0.9) | 57.39 | 15.67 | 35.98 | 34.27 | 35.83 | 1.81 |
| | 11 | LoRAQuant (3@0.8) | 60.05 | **18.03** | 35.37 | 34.74 | 37.05 | 2.17 |
| | 12 | LoRAQuant (3@0.9) | **60.8** | 17.44 | **39.02** | **34.95** | **38.06** | 2.5 |

Table 1: Performance and average bitwidth for different methods. "Avg Perf." refers to average performance. In Rows 5–12, we **bold** the best task performance, as well as the least average bit among models.

rized and others maintained in full precision; note that an additional indicator bit is needed for each weight. **BiLLM** (Huang et al., 2024) works similarly, except that each column must be either quantized or maintained in full precision while using a split binarization strategy. All baselines perform group quantization with a group size of 128, which is a common practice in the literature (Lin et al., 2024; Frantar et al., 2023).

We also consider **JD-Diagonal** (Gabrielsson et al., 2024), which is not a quantization method but is related to the general objective of our research: reducing memory for multiple LoRAs. Specifically, JD-Diagonal clusters different LoRAs and performs weight sharing, while introducing a few ($r$-many) additional parameters for each task. In our evaluation, we treat the three tasks as a cluster for weight sharing.

Our comparison focuses on two aspects: (1) output quality measured by the standard metric of each task, and (2) average number of bits per LoRA parameter,[3] which reflects the effectiveness of memory saving. When computing the average bits, we also consider the scale and the zero point parameter in our computation. The values reported in Tab. 1 represent the average bitwidth across the three task-specific adapters. For detailed per-adapter bitwidth, refer to Appendix C.

---

[3]Notice that our paper focuses on LLM customization where massive numbers of LoRAs are loaded to a base, high-performing LLM. Therefore, our base LLM follows a standard QLoRA treatment and its parameter size (which is a constant) is not considered in the metric. Appendix D provides a memory analysis with the base LLM.

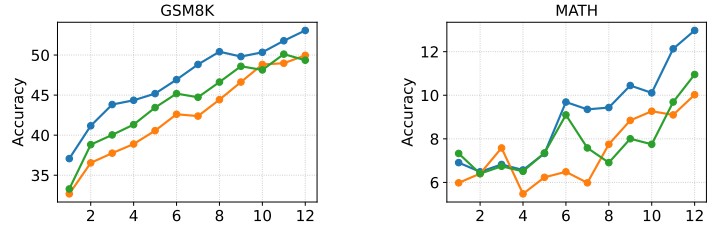

Figure 2: Comparison of sub-LoRA splitting strategies. Here, $h$ denotes the rank of the high-precision sub-LoRA and is fixed globally for all LoRAs in a model.

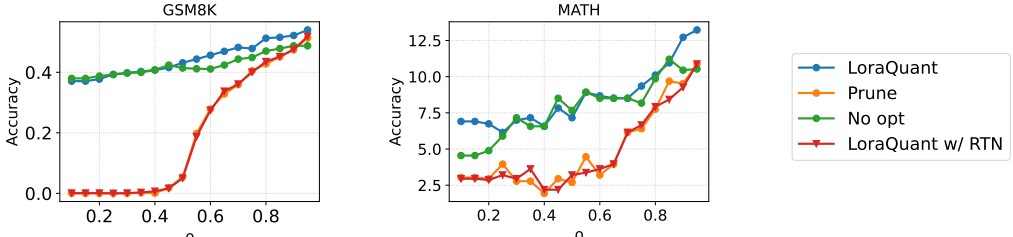

Figure 3: Study on optimization and quantization of LORAQUANT. **LoraQuant** is the proposed method. **Prune** truncates the less important sub-LoRA components. **No Opt** omits the gradient-based optimization step. **LoraQuant w/ RTN** replaces the specialized binarization with 1-bit RTN quantization.

As shown in Tab. 1, FP16 (Row 1) achieves the highest overall performance across all models, which is expected. Various quantization methods (Rows 2–3 and Rows 5–8) provide a spectrum of performance–memory tradeoff. However, all previous methods require at least two bits to achieve reasonable performance. As we see, binary quantization (Row 2) yields 30-point degradation in accuracy on GSM8K with LLaMA 2-7B, whereas RTN (1 bit) almost collapses the model, leading to extremely poor accuracy.

We also observe that JD-Diagonal fails to achieve reasonable performance in our experiments. We hypothesize that the discrepancy between our experiment and that in Gabrielsson et al. (2024) is due to the tasks and evaluation metric. Gabrielsson et al. (2024) only consider simple tasks with in-context learning samples, which can already be handled well by the base model. In our pilot study, we find that even reducing the LoRA rank to 1 via SVD has little impact on the performance of their trained adapters, suggesting that the LoRA parameters contribute minimally in their settings. Also, they only use ROUGE-L as their metric, while in our setting math and coding require exact matching.

We then examine our LORAQUANT approach (Rows 9–12). LORAQUANT performs mixed-precision quantization, where the more important sub-LoRA is quantized to $i$ bits ($i = 2$ or $3$), and the less important sub-LoRA is always quantized to 1 bit. The fraction is controlled by the hyperparameter $\rho$, which is the total variation explained (§3.1). Our variant is denoted by LO-RAQUANT($i@\rho$).

As seen, the 2@0.8 and 2@0.9 variants (Rows 9–10) consistently operate under 2 bits, while achieving high performance comparable to, or even higher than, PB-LLM and BiLLM, which are also mixed-precision quantization but use more bits than ours. To enable a fair comparison with these binarization methods at a similar average bitwidth, we also experiment with quantizing the more important sub-LoRA to 3 bits. Our 3@0.8 and 3@0.9 variants (Rows 11–12) consistently achieve higher task performance than both PB-LLM and BiLLM.

To conclude, our LORAQUANT method is able to reduce the memory of LoRAs by a large margin while achieving comparable performance with full precision baseline. Our approach also largely advances extreme quantization with less than two bits per parameter in the LoRA setting.

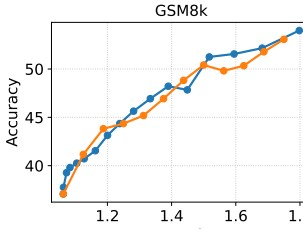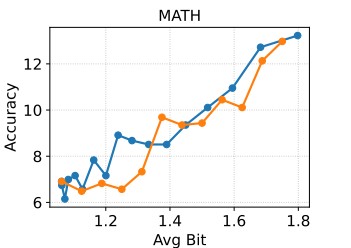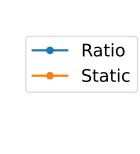

Figure 4: Comparison of $h$ selection strategy. Ratio denotes our method explained in §3.1, where the ratio hyperparameter varies from 0.1 to 0.95 in increments of 0.05, while Static sets $h$ to a fixed value ranging from 1 to 12.

### 4.3 IN-DEPTH ANALYSES

We conduct in-depth analyses on the LLaMA-2-7B model using the GSM8K and MATH datasets. We restrict our analyses to this setting due to computational constraints.

**Sub-LoRA split strategy.** In Fig. 2, we compare splitting sub-LoRAs using SVD against two baseline strategies: (i) selecting columns of $\mathbf{B}$ and the corresponding rows of $\mathbf{A}$ at random, and (ii) selecting the high-precision components according to the magnitude of $\mathbf{b}_i \mathbf{a}_i$ (where $\mathbf{b}_i$ denotes the $i$th column of $\mathbf{B}$ and $\mathbf{a}_i$ denotes the $i$th row of $\mathbf{A}$), measured by its Frobenius norm. The intuition behind the norm-based strategy is that components with larger norms contribute more substantially to the overall LoRA update, and thus are more suitable for higher-precision quantization. In this analysis, we do not choose $h$ dynamically (§3.1) but directly set the value of $h$ to ensure a fair comparison with the random and norm-based strategies. As seen in the plots, our SVD split strategy generally outperforms the other methods. This is consistent with our intuition that SVD identifies the important dimensions, for which we should preserve more information during quantization.

**Ablation study.** In Fig. 3, we analyze the effect of different components of our approach. First, we ablate the gradient-based optimization, which searches for reparameterization (after splitting the sub-LoRAs) to reduce quantization error (§3.3). Fig. 3 shows that the optimization generally improves the performance of LORAQUANT, with higher improvement for higher ratios; therefore, we adopt the optimization process in our approach.

Next, we ablate our approach by pruning the less important sub-LoRA entirely to test whether it contributes to performance. As shown in Fig. 3, pruning collapses the model at lower ratios, which is expected. As the ratio increases, the performance of pruning also increases, but is consistently lower than our LORAQUANT. This verifies that the less important sub-LoRA, even quantized to 1 bit per weight, still plays a role in the model.

We also experiment an alternative variant of LORAQUANT, where the less important sub-LoRA is quantized by 1-bit RTN, instead of sign-based binarization (§3.2). This setting performs similarly to pruning the less important sub-LoRA, as 1-bit RTN effectively maps lots of values to zero. The result justifies our different quantization methods used: RTN for the important sub-LoRA and sign-based binary quantization for the less important one.

In Fig. 4, we compare our ratio-based dynamic $h$ selection with a static strategy that fixes $h$ as a global value. The results show that a dynamic $h$ generally yields better performance, particularly when we allow for slightly more bits (e.g., more than 1.5 bits), which is a more practical setup.

## 5 CONCLUSION

In this paper, we address the problem of memory reduction when multiple LoRAs are loaded simultaneously in the scenario of LLM customization (e.g., for different tasks and/or users). We propose LORAQUANT, a mixed-precision quantization method tailored for LoRAs, where we split each LoRA into two sub-LoRAs via SVD. The more important sub-LoRA is preserved with more bitwidth, whereas the less important sub-LoRA is quantized to one bit; we further adopt straight-through gradient optimization to improve the quantization. Experimental results demonstrate that LORAQUANT maintains strong performance even under extremely low bitwidth settings. We further present in-depth analyses to verify the effectiveness of each component in our approach.

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

## A  IMPLEMENTATION DETAILS

We follow the hyperparameter settings of Biderman et al. (2024), except that we reduce the batch size because they train across multiple GPUs while we only use a single GPU.

- **optimizer:** adamw_torch ($\beta = [0.9, 0.95]$)
- **learning rate:** $2 \times 10^{-4}$
- **scheduler:** cosine_with_warmup ($\alpha_f = 0.01$, $t_{\text{warmup}} = 0.3 \, \text{dur}$)
- **weight decay:** 0
- **precision:** fp16
- **device_train_microbatch_size:** 6
- **gradient clipping:** norm (threshold $= 1$)
- **num_epochs:** 2
- **num_gpus:** 1

For both mathematical reasoning and summarization tasks, we set **max_seq_len** to 1024. For the code domain, we use **max_seq_len** $= 4096$. The **batch_size** is 16 for Mistral and LLaMA-2-7B, and 8 for LLaMA-2-13B.

## B  QUANTIZING ALONG COLUMN OR ROW

In our approach, we perform group quantization, where $\mathbf{B}'$ is quantized column-wise and $\mathbf{A}'$ row-wise. This follows naturally from the SVD reparameterization of the adapter weight: the square root of each singular value is multiplied with the corresponding columns of $\mathbf{U}$ to form $\mathbf{B}'$, and rows of $\mathbf{V}^\top$ to form $\mathbf{A}'$. Under this scheme, the singular values can be absorbed into the RTN scaling factors stored in FP16, ensuring that the singular values are preserved without error.

We also explore alternative quantization strategies. Our hypothesis is that column-wise quantization of $\mathbf{B}'$ and row-wise quantization of $\mathbf{A}'$ should yield stronger performance. The results of this comparison are presented in Fig. 5. As shown, this hypothesis holds for the GSM8K dataset, where this configuration performs best, but on the MATH dataset, no singular strategy wins consistently. That being said, the performance difference remains small, and we adopt column-wise quantization of $\mathbf{B}'$ and row-wise quantization of $\mathbf{A}'$ as the default setting in our approach since it makes more sense intuitively.

## C  AVERAGE BITWIDTH OF LORAQUANT

In our work, we use AvgBits to measure the memory usage, given by

$$\text{AvgBits} = \frac{\text{total bits for LoRAs across different layers}}{\text{total \# of LoRA parameters across different layers}}. \tag{10}$$

In the main experiment (Tab. 1), we present the average bitwidth across multiple tasks, due to table formatting reasons. In this appendix, we show the bitwidth for individual tasks in Tab. 2. Notice that we have a dynamic allocation strategy, so the average bits vary slightly based on the task.

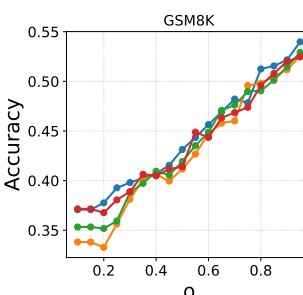 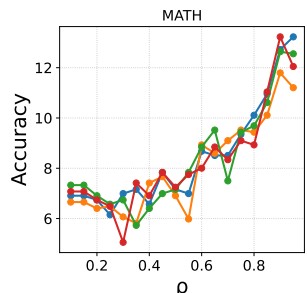 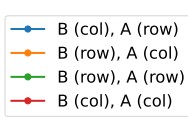

Figure 5: Study on the column-wise and row-wise quantization of LORAQUANT. Each entry is denoted as **B** (_) **A** (_), where each underscore can be either **col** or **row**. Here, **col** indicates column-wise quantization and **row** indicates row-wise quantization of the corresponding component.

| Model | Method | GSM8k & MATH | HumanEval | Xsum |
|---|---|---|---|---|
| LLaMA 2-7B | LORAQUANT (2@0.8) | 1.65 | 1.55 | 1.74 |
| | LORAQUANT (2@0.9) | 1.82 | 1.74 | 1.89 |
| | LORAQUANT (3@0.8) | 2.17 | 1.98 | 2.34 |
| | LORAQUANT (3@0.9) | 2.51 | 2.34 | 2.65 |
| Mistral 7B | LORAQUANT (2@0.8) | 1.86 | 1.82 | 1.85 |
| | LORAQUANT (2@0.9) | 1.98 | 1.95 | 1.97 |
| | LORAQUANT (3@0.8) | 2.59 | 2.51 | 2.58 |
| | LORAQUANT (3@0.9) | 2.83 | 2.76 | 2.81 |
| LLaMA 2-13B | LORAQUANT (2@0.8) | 1.61 | 1.56 | 1.77 |
| | LORAQUANT (2@0.9) | 1.79 | 1.75 | 1.91 |
| | LORAQUANT (3@0.8) | 2.09 | 2.00 | 2.40 |
| | LORAQUANT (3@0.9) | 2.44 | 2.37 | 2.69 |

Table 2: Average bitwidth of LORAQUANT variants.

# D   MEMORY ANALYSIS FOR LLM CUSTOMIZATION

In this appendix, we provide an analysis of memory savings when loading different numbers of adapters, where we use the average bitwidth reported in the main experiments (specifically, the 2@0.8 setup on the GSM8K dataset in Tab. 1). As shown in Fig. 6, the memory usage of FP16 grows substantially when the number of LoRAs increases. For instance, loading only 50 LoRAs requires 2 times more memory than the base LLM. This justifies our motivation that the memory of multiple LoRAs can build up and become non-negligible, even if each LoRA is relatively lightweight. On the contrary, the memory grows slowly in our LORAQUANT method, showing its practical values for LLM customization.

# E   LIMITATIONS AND FUTURE WORK

Our paper proposes a novel LoRA quantization method for LLM customization. In the experiments, we have tried four datasets and three language models (12 evaluations in total). Due to the limit of our computing resources, we are unable to perform commercial scale experiments (e.g., having millions of LoRAs). Luckily, our method treats different LoRAs independently, and thus is easily scalable, as opposed to Gabrielsson et al. (2024) who require recomputing cluster parameters. Appendix D has shown the practival values of our model, and we leave the commercial applications of our method to the industry.

Moreover, our SVD-based mixed-precision quantization is specific to LoRA models and is not directly applicable to a full weight matrix. This is because, if a matrix is not low-rank, splitting a matrix into two products of sub-matrices would in fact increase the number of parameters. In the

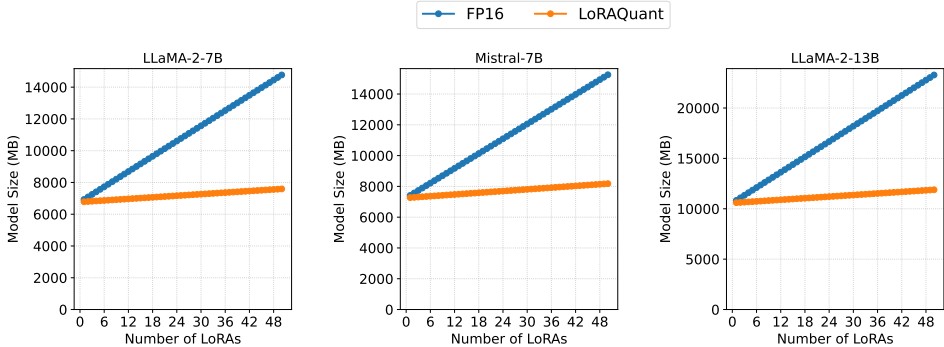

Figure 6: Memory usage when loading multiple LoRAs and the base LLM.

future work, we plan to adapt our method to general weight matrices by truncating the SVD dimensions. However, this is clearly beyond the scope of this paper. That being said, our LORAQUANT approach can be combined with other quantization methods, and in our experiments, we have already used a four-bit quantized base models.

## F REPRODUCIBILITY STATEMENT

The code for running the experiments and training adapters is released as an anonymous repository (Footnote 1). The datasets used for training and evaluation are all publicly available.

## G LLM USAGE DISCLOSURE

ChatGPT5 was used in a limited capacity to check grammar, rephrase certain expressions, and help format the tables in LaTeX and figures in Matplotlib. However, we came up with the research ideas, conducted the analyses, and presented the contents without using AI tools.

