# OpenReview forum: "LoraQuant: Mixed-Precision Quantization of LoRA to Ultra-Low Bits"
_ICLR.cc/2026/Conference — ICLR 2026 Conference Withdrawn Submission_

### Official Review · Reviewer_c4UH · 2025-10-27

**Soundness:** 3
**Presentation:** 3
**Contribution:** 2
**Rating:** 2
**Confidence:** 4

**Summary:**

This paper introduces LoraQuant, a mixed-precision post-training quantization method specifically tailored for Low-Rank Adapters (LoRA) used in large language model (LLM) customization. The key idea is to apply SVD-based decomposition to separate LoRA components into important and less important sub-LoRAs, quantizing the former at slightly higher precision (2–3 bits) and the latter at ultra-low (1-bit) precision. The authors claim this approach maintains competitive accuracy at sub-2-bit average precision across multiple LLMs (LLaMA2-7B/13B and Mistral 7B) and tasks.

**Strengths:**

- Clear motivation: Addresses the practical issue of memory overhead when serving multiple LoRAs simultaneously.

- Systematic evaluation: The experiments cover several models and tasks, with quantitative comparisons against GPTQ, PB-LLM, and BiLLM.

- Empirical rigor: The ablation studies are comprehensive, isolating the impact of SVD splitting, optimization, and dynamic precision allocation.

**Weaknesses:**

- Limited novelty: The main technical components, SVD, mixed-precision quantization, and straight-through optimization, are all well-established techniques in the quantization and model compression literature.

  - Similar SVD-based quantization (e.g., SVDQuant, PiSSA) already exist.

  - Mixed-precision binarization has been explored in PB-LLM and BiLLM.

  - Using SVD to rank component importance is standard practice in low-rank adaptation.

- Narrow scope of contribution: The method applies known ideas to a specific setting (LoRA modules) without fundamentally advancing the theory or algorithms of quantization.

- Incremental empirical gains: Although the reported performance is competitive, improvements over recent baselines (e.g., BiLLM, GPTQ) are modest, suggesting a refinement rather than a breakthrough.

- Lack of new insights: The paper mostly reuses the established quantization pipeline and adapts it to LoRA’s low-rank form. There is limited theoretical or analytical contribution to understanding why SVD-splitting benefits quantization beyond empirical observation.

**Questions:**

- Could the authors clarify whether the proposed method offers any measurable benefits in terms of inference speed, latency reduction, or throughput improvement?

---

### Official Review · Reviewer_UvxM · 2025-10-29

**Soundness:** 2
**Presentation:** 2
**Contribution:** 2
**Rating:** 4
**Confidence:** 4

**Summary:**

This paper proposes LoraQuant, a mixed-precision post-training quantization method for LoRA adapters. LoraQuant reparameterizes each LoRA (BA) via singular value decomposition (SVD), enabling it to split the adapter into two sub-LoRAs: high-precision part and ultra-low-bit (1-bit) part. It further applies a straight-through gradient-based optimization to minimize quantization error. Experiments on LLaMA2-7B, LLaMA2-13B, and Mistral-7B across GSM8K, MATH, HumanEval, and XSum show that LoraQuant achieves competitive accuracy under <2 bits on average, outperforming baselines like GPTQ, PB-LLM, and BiLLM in the extreme low-bit regime.

**Strengths:**

1. Instead of applying generic quantization, LoraQuant propose to reparameterize LoRA through SVD, leveraging the inherent low-rank structure to guide mixed-precision assignment.
2. The proposed LoraQuant method reparameterizes $BA$ into $US^{1/2}$, $S^{1/2}V^T$, where the singular values naturally encode the importance of each latent direction.
2. LoraQuant achieves competitive or superior accuracy under <2 bits on average, significantly outperforming standard baselines such as RTN, GPTQ, PB-LLM, and BiLLM in extreme low-bit regimes.

**Weaknesses:**

1. While the paper focuses on compressing LoRA adapters, it is unclear how significant the overall memory savings are when the base model remains dominant. For example, when the LoRA rank is 64, the adapter typically constitutes only about 2–3% of the base model parameters. In such cases, quantizing only the LoRA part may have marginal benefits compared to compressing or quantizing the base model itself to an extreme degree.
2. The proposed straight-through optimization requires about 100 steps for each row or column of the LoRA matrices. For a LoRA with $m$ rows, this implies roughly $100×m$ optimization steps per adapter. However, the paper does not provide any runtime analysis to quantify the computational cost time of this process.
3. Including results on additional models (e.g., LLaMA3) would strengthen this paper.
4. LoraQuant is applied after training a full-precision LoRA adapter on a quantized base model (using QLoRA). However, QLoRA itself is not the most optimal quantization-aware fine-tuning framework. More recent methods such as ApiQ or CLoQ explicitly co-optimize adapter parameters under quantization constraints and could yield stronger baselines. Comparing against or integrating with these methods would make the evaluation more convincing.

[1] Liao, Baohao, et al. "Apiq: Finetuning of 2-bit quantized large language model." arXiv preprint arXiv:2402.05147 (2024).

[2] Deng, Yanxia, et al. "Cloq: Enhancing fine-tuning of quantized llms via calibrated lora initialization." arXiv preprint arXiv:2501.18475 (2025).

**Questions:**

1. Have you tried quantizing more than two sub-LoRAs? Why just two sub-LoRAs?
2. Can you provide the computational cost time of optimization steps?
3. Can the SVD reparameterization and optimization be integrated into LoRA training (not only post-training)?

---

### Official Review · Reviewer_TzzB · 2025-10-30

**Soundness:** 2
**Presentation:** 3
**Contribution:** 1
**Rating:** 2
**Confidence:** 5

**Summary:**

This paper proposes quantizing LoRA weights to reduce memory usage when deploying multiple LoRA models in practice. Specifically, it applies Singular Value Decomposition (SVD) to each LoRA weight matrix and then uses mixed precision—keeping important singular directions in higher precision and less important ones in 1-bit precision. The methods are evaluated on Llama2 and Mistral models across several popular downstream tasks.

**Strengths:**

1). The paper is clearly written and easy to follow.

2). The ablation studies are clear and well presented.

**Weaknesses:**

1). The technical contribution has limited novelty. Essentially, the paper applies the same algorithm as SVDQuant to LoRA weights. Other quantization techniques mentioned, such as 1-bit or 2-bit quantizers and optimizations for reducing quantization error, are standard.

2). The evaluation misses an important baseline—SVDQuant.

3). The performance improvements are limited, especially compared to other quantization methods with similar average bits.

4). The evaluation is conducted only on relatively older model versions. It should at least include the latest Llama and other popular open-source models.

5). Details of the optimization process are missing. The paper only mentions “within one hundred gradient steps.” It does not specify what dataset was used, how much data was involved, or other relevant setup details.

**Questions:**

1). What are the inference-time overheads of using this SVD + mixed-precision approach, in terms of both latency and peak memory usage?

2). Is the optimization tuning also applied to other quantization baselines listed in Table 1?

---

### Official Review · Reviewer_DqRW · 2025-10-31

**Soundness:** 3
**Presentation:** 3
**Contribution:** 2
**Rating:** 2
**Confidence:** 5

**Summary:**

The paper introduces LORAQUANT, a mixed-precision quantization method for LoRA adapters in large language models. Motivated by multi-task or multi-user settings where many LoRAs may be loaded simultaneously, it decomposes each LoRA via SVD to separate important and less important components. The former are quantized with 2–3 bits, the latter with 1 bit, followed by a small optimization step to reduce quantization error. Experiments on LLMs are conducted.

**Strengths:**

1. Clear motivation and simple SVD-based mixed-precision design.
2. Strong low-bit results with minimal accuracy loss.
3. Comprehensive experiments and ablations across multiple LLMs and tasks.

**Weaknesses:**

1. LoRA adapters are already small, so quantizing them offers little real-world benefit. For the multi-Lora case, usually the Lora adapters can be saved offline and loaded during inference. The proposed work can be only useful when multiple Lora adapters have to be uploaded simultaneously at the same time. It would be better for authors to discover more on where the proposed approach can have an impact on.
2. In the scenario that multiple Lora adapters have to be loaded simultaneously, there can be many easy ways to get compact Lora adapters. For instance, directly fine tune with quantization-aware training to get low-precision adapters or finetune using lower ranks. These methods are likely to yield higher accuracy than post-training quantization. The paper should include a comparison or discussion of these alternatives.
3. There still exists a gap between the compressed LoRA and the full-precision ones. It further increases the gap from the full-finetuning results. Also the improvement seems marginal especially compared with BiLLM which also uses mixed-precision strategy.
4. In addition, the GPTQ was proposed for the problem $\min_Q \|XW-XQ\|^2$, in your experiment, how do you use GPTQ to quantize A and B?
5. Although the paper’s motivation centers on multi-LoRA or multi-user deployment, the experiments only quantify accuracy. There are no measurements of real GPU memory savings, inference throughput, or latency when loading many adapters concurrently
6. As acknowledged by the authors, similar SVD-based mixed-precision quantization strategies have already been explored in prior works such as SVDQuant and related low-rank quantization methods. Therefore, the novelty of this paper is limited, as it mainly adapts an existing concept to LoRA adapters rather than introducing a fundamentally new quantization technique, specifically for LoRA adapters.

**Questions:**

See weakness.

---

### Note · Authors · 2025-11-25

I have read and agree with the venue's withdrawal policy on behalf of myself and my co-authors.